# Experimental Study and Modeling of the Fracture Behavior, Mechanical Properties, and Bonding Strength of Oil Well Cement

**Cumaraswamy Vipulanandan [1,\*], Ahmed Salih Mohammed [2,3,\*]** and **Praveen Ramanathan [4]**

1    Center for Innovative Grouting Materials and Technology (CIGMAT), Department of Civil and Environmental Engineering, University of Houston, Houston, TX 77204, USA
2    Department of Civil Engineering, College of Engineering, University of Sulaimani, Sulaimaniyah 46001, Kurdistan Region, Iraq
3    Engineering Department, Civil Engineering, American University of Iraq, Sulaimani (AUIS), Sulaimaniyah 46001, Kurdistan Region, Iraq
4    Department of Civil and Environmental Engineering, University of Houston, Houston, TX 77204, USA
\*    Correspondence: cvipulan@central.uh.edu or cvipulanandan@uh.edu (C.V.); ahmed.mohammed@univsul.edu.iq (A.S.M.); Tel.: +964-770158865 (A.S.M.)

**Abstract:** This study aimed to analyze the outcomes of stress intensity factor ($K_I$) and new bond strength tests of oil well cement (class H) with a water-to-cement ratio (w/c) of 0.38. Mechanical properties of the cement paste, such as the compressive and flexural strengths, were tested and qualified at 1, 7, and 28 days of curing. The relationship between the elastic modulus and axial strain using the differential of the Vipulanandan p-q model for the cement paste was obtained. The stress intensity factor of the cement paste was between 0.3 and 0.6 MPa.m, and the crack tip opening displacement (CTOD) was between 2.798 and 6.254 μm at three different ratios between the initial notch height (a) and the thickness of the beam (d) (a/d = 0.3, 0.4, and 0.5). The nonlinear Vipulanandan p-q model was used to model the compressive and flexural stress–strain behavior of the cement at three curing times. The bonding strength between the cement and steel tube representing the casing in the borehole was 0.75, 1.89, and 2.59 MPa at 1, 7, and 28 days respectively.

**Keywords:** mechanical properties; fracture mechanics; curing age; bonding strength; models

## 1. Introduction

Each cement particle forms a type of growth on its surface during hydration. It gradually spreads until it adheres to the growth of other adjacent cement particles, resulting in progressive stiffening, hardening, and strength development [1]. Even if cement is a well-consolidated material, the chemistry of cement (and the chemistry inside cement) remains very complex and nonobvious. What is certain is that the hydration mechanism plays a pivotal role in the development of cement with specific final chemical compositions, mechanical properties, and porosities. This document provides a survey of the chemistry behind such inorganic materials. The text has been organized into five parts describing: (i) the manufacturing process of Portland cement, (ii) the chemical composition and hydration reactions involving Portland cement, (iii) the mechanisms of setting, (iv) the classification of the different types of porosities available in cement (with particular attention given to the role of water in driving the formation of pores), and (v) the recent findings on the use of recycled waste materials in cementitious matrices (with a particular focus on the sustainable development of cementitious formulations). In this study, the influence of water on the main relevant chemical transformations occurring in cement emerged with the formation of specific intermediates/products that might affect the final chemical composition of types of cement. Within the text, a clear distinction between setting and hardening is provided. Water's physical/structural role in influencing the porosities in cement is analyzed, providing a correlation between types of bound water and porosities [2,3]. In the petroleum industry, cement has been used in oil well operations. Cement

is typically utilized to fill the annular space between the casing and rock formation by displacing the drilling fluid. In addition, cement will support the casing and protect it against corrosion and impact loading, restrict the movement of fluids between formations, and isolate productive and nonproductive zones. Oil well cement is used under different exposure conditions than the cement used in the conventional construction industry. The strength of oil well cement usually depends on factors such as time and conditions of curing, environmental conditions, slurry design and use of additives, and any additional treatments to the cement [4].

Cementing primarily aims to completely and irreversibly isolate the formation behind the casing. The success of a production operation of a well is almost entirely predicated on the quality of the primary cementing work. A full hydraulic seal is present between the formation and the casing across all the zones of interest [3]. Cementing also creates a hydraulic seal, which stops fluids from moving between producing zones in the borehole and escaping to the surface. In addition, it prevents the steel casing from rusting when exposed to formation fluids [5]. Over time, the cement is subjected to stresses due to pressure integrity tests, an increase in mud weight, casing perforation, well stimulation, the production of oil or gas, and an increase or fluctuation in the wellbore temperature [3,6]. Recent case studies on cementing damages found several problems that have caused varied cementing process delays [2,3]. These problems caused the cementing procedures to run longer than expected. Controlling fluid loss to rock and soil formation and effective well cementing have become essential challenges in oil well construction to maintain wellbore integrity [6]. This is because changing downhole environments threatens the integrity of the wellbore. Consequently, comprehensive monitoring and control of the entire oil-well-cementing process are necessary to ensure the cement maintains its integrity throughout its useful life [7,8].

The safety and effectiveness of the $CO_2$ injection procedure for geologic carbon storage depend on the integrity of the cement, which offers zonal isolation and mechanical support. This research focused on radial cracking in cement after $CO_2$ injection and interfacial debonding at wellbore contacts. It applied the energy release rate (ERR) definition to describe how fractures spread. The suggested model used the finite element approach and calculated the ERRs of both fractures using realistic wellbore layouts and injection settings. Additional parametric research revealed how the fracture geometry, the cement's mechanical and thermal characteristics, and the crack size affected crack propagation. According to simulation findings, with normal cement characteristics, interfacial and radial fracture ERRs would be more than 100 J/m². The Young's modulus, Poisson's ratio, and thermal conductivity of the cement were the next most significant influences on the ERR. Other crucial factors in regulating fracture propagation are the cracks' starting sizes and locations.

Furthermore, nonuniform in situ loads would accelerate fracture propagation at the interfaces. These important results might contribute to the improvement of cement sheath design and guarantee the long-term integrity of wells used for geological carbon storage. [1,9,10]: (i) cracking the cement sheath could allow fluids to migrate radially and vertically; and (ii) plastic deformation in the cement sheath could also allow fluids to migrate radially and vertically (Figures 1 and 2). A cement or concrete fracture manifests as cracking as an indication of internal stress or deterioration. Cement and concrete cracking is common, but they should never occur and usually mean more significant problems. Cement and concrete crack and break down into their respective stages, and cracking or fracturing in cement and concrete occur in three distinct phases. It is essential to distinguish between the mode of crack initiation and how this occurs at the microscopic level, the subsequent paths of propagation, and the ultimate macroscopic crack pattern. Strain and stress concentrations resulting from the incompatibility of the aggregate and paste components' elastic moduli are primarily responsible for forming small fissures or microcracks in fresh, hardened cement and concrete. The following are the different phases of concrete and cement cracking. In the first step, strain concentrations at the aggregate–

paste interface can occur before loading due to shrinkage or thermal movements in the cement or concrete [3,11,12].

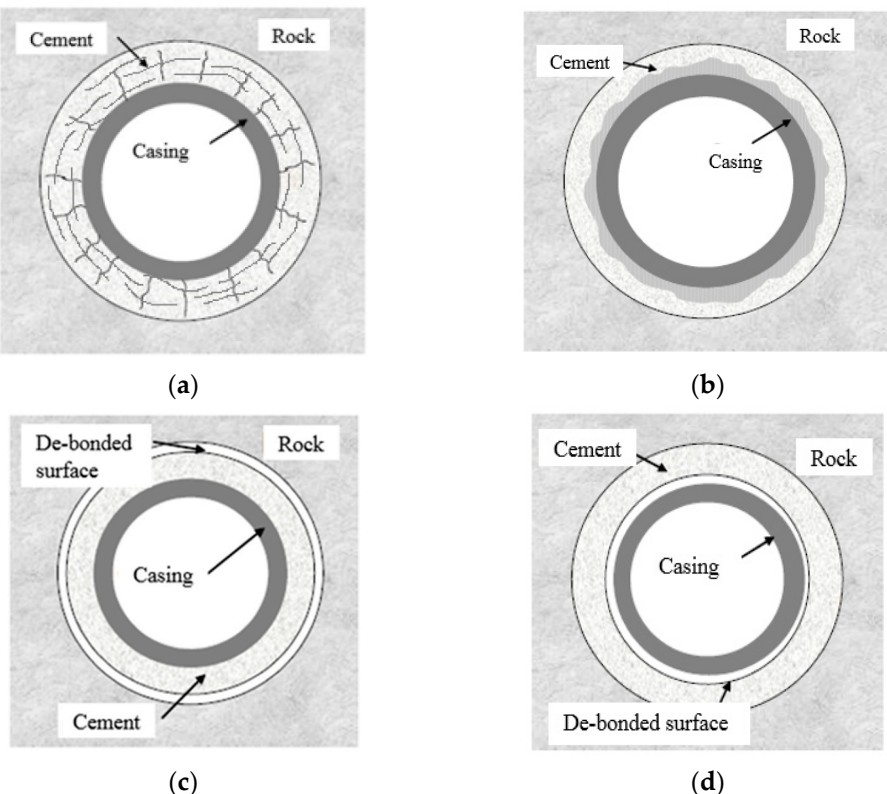

**Figure 1.** Debonding at Steel–Cement Interface. (**a**) Cracks in Cement Sheath, (**b**) Plastic Deformation in Cement Sheath, (**c**) De-bonding at Rock–Cement Interface, and (**d**) Debonding at Casing–Cement Interface [13].

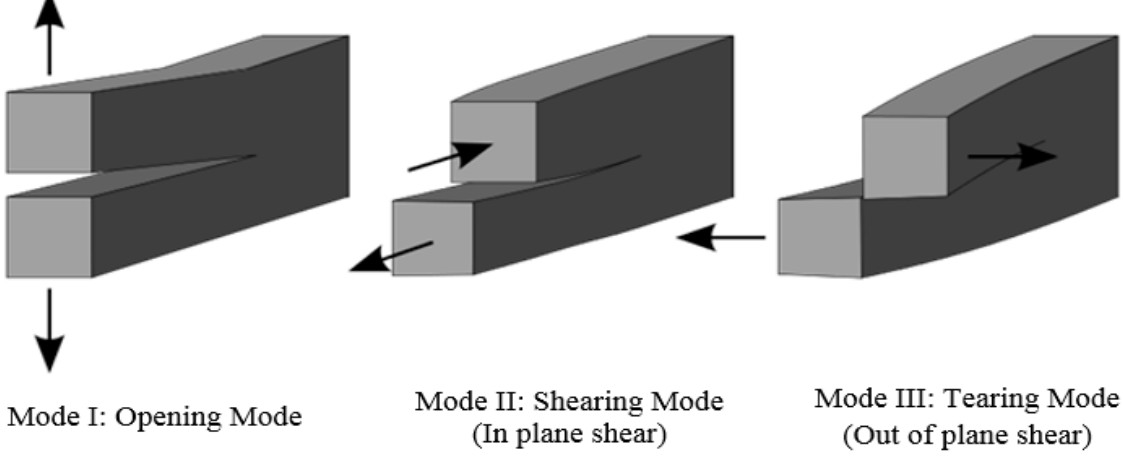

**Figure 2.** Fracture Modes of Materials. Tearing Mode [13].

In Phase I, cracks start microscopically at specific locations all over the specimen where the tensile strain is the highest. The lack of further crack growth at this load level demonstrates the stability of the cracks. Once the load is increased, initially stable cracks begin propagating in Phase II. Due to the likelihood of overlap between stable crack initiation and crack propagation, the transition between Stages I and II will be gradual (see Figure 2 for a visual demonstration). In Stage II, the crack system multiplies and spreads slowly and steadily, so propagation stops if loading stops and the stress level is

constant [13]. The pressures acting from both sides of the sheath, the outside formation pore pressure, and the inside pressure transmitted from the casing and other axial loadings are the causes of the stresses induced in wellbore cement. Axial loadings can cause these stresses [14].

In most cases, it causes the material to transform into a different shape. The cement sheath can debond at the following interfaces [11]: cracking of the cement sheath, which could allow radial and vertical migration of fluids (Figure 1a); plastic deformation in the cement sheath, which could allow radial and vertical migration of fluids (Figure 1b); the rock–cement interface (Figure 1c); and the cement–casing interface (Figure 1d). In addition, the pressure and thermal cycle damage the cement sheath's integrity (sealing integrity and mechanical integrity), leading to gas leakage. The failure mode of a downhole cement sheath can be divided into four categories: (1) debonding/micro annulus at the interface of the cement sheath, (2) shear/tensile failure of the cement sheath, (3) radial cracks of cement sheath, and (4) plastic deformation of the cement sheath [10,13]. Researchers have used Linear Elastic Fracture Mechanics (LEFM) parameters and Elastic-Plastic Fracture Mechanics (EPFM) parameters in the literature to characterize the fracture resistance of cement paste, mortar, cement concrete, polymer concrete, and rocks. Figure 2 shows that the fracture mechanics field considers three distinct crack propagation modes. Mode I, the opening crack propagation model, is caused by a tensile stress field. The shearing or sliding mode that results from in-plane shear is Mode II. Mode III, also known as the tearing mode, is caused by shear outside the material's plane [15]. In engineering practice, the stress state near the tip of a sharp crack is more useful. In a linear elastic material, the stress intensity factor ($K_I$) characterizes the crack tip conditions [13–15]. If $K_I$ is known, simple equations can be used to determine how the stress is distributed at the crack tip. No study was conducted on evaluating the fracture behavior of oil well cement at different heights of the initial crack.

Vipulanandan, C., and Dharmarajan [15,16] studied the Mode I fracture regarding the applicability of the critical stress intensity factor ($K_{IC}$) and/or critical crack-up opening displacement ($CTOD_c$) for epoxy and polyester polymer composites (PCs). Single-edge notched beams were used to study unreinforced and glass-fiber-reinforced polymer composite systems at room temperature in three-point and four-point bending. The crack extension during the pre-peak stress was calculated using the crack mouth opening displacement (CMOD) technique. The effective crack's tip was determined by the critical stress intensity factor, and the elastic crack tip opening displacement served as a proxy for the critical crack extension. The performance of this model was compared to other models provided for metals and cement concrete to calculate the elastic CTOD from the observed elastic CMOD. The test findings showed that the initial notch depth did not affect the two fracture characteristics. The two fracture characteristics predicted the notch sensitivity of polyester and epoxy polymer concrete. Resistance curves were created for a 4% glass-fiber-reinforced polyester PC based on the stress intensity factor and crack tip opening displacement [16]. Feng et al. [17] studied and thoroughly characterized the fracture mechanics of Portland cement mortars reinforced with multiwall carbon nanotubes and carbon nanofibers. To strengthen cement mortars with well-dispersed carbon nanotubes and carbon nanofibers, the critical values of the stress intensity factor, strain energy release rate, crack tip opening displacement, and critical crack length were experimentally found. A three-point closed-loop bending test was performed on prismatic notched specimens of neat mortars, mortars reinforced with 0.1 weight percent carbon nanofibers, and mortars reinforced with 0.1 and 0.2 weight percent multiwall carbon nanotubes. The crack mouth opening displacement was used as the feedback signal. The two-parameter fracture model was then used to calculate the fracture parameters of the nano-reinforced mortars [18]. The results of assessing parameters like compressive strength, modulus of elasticity, compressive toughness, flexural strength, flexural toughness, flexural residual strength, and fracture energy on steel and polypropylene fiber reinforced concrete that will be used in industrial ground floor slabs. The characterization approach included creating

cylindrical and prismatic specimens of fiber-reinforced concrete with nine different kinds of fibers, including four synthetic or polypropylene and five steel fibers. On the basis of a design for concrete with a 40 MPa flexural strength used for industrial ground floor slabs, non-reinforced concrete samples were also mixed as a reference. Based on the findings, it was concluded that the length and shape of fibers have an impact on the development of flexural strength as well as some other mechanical properties of concrete, such as ductility, since concrete samples made with steel fibers with hooks at their extremes exhibited better adhesion to concrete, while straight copolymer fiber and wavy polymer fiber displayed better overall performance. Finally, it was shown that the values of compression toughness and flexural toughness exhibited a strong association [19].

Cracks are commonly found in cement and concrete structures. They are undesirable features that may be brought about by environmental causes, workmanship, natural causes, as well as the age of the concrete element. It is, therefore, essential that the causes and consequences of cracking are well understood so that suitable remedial measures can be adopted. Once the cement sheath seal fails, it will lead to issues such as fluid leakage and sustained casing pressure. Cyclic loading will cause cumulative plastic strain and strength degradation of cement stone, a relatively dangerous working condition for the downhole cement sheath.

This study evaluated and modeled the mechanical behaviors of oil well cement, such as the stress intensity factor, compressing, and flexural stress–strain under cycling load.

*Research Significance*

This study aimed to assess the mechanical performance of an oil well cement, including its bonding, compressive, and flexural strengths. Among the specific objectives were the following:

i.     Identifying the mechanical properties of oil well cement at various curing times.
ii.    Researching Mode I and evaluating the fracture properties of oil well cement using three different a/d ratios.
iii.   Using the nonlinear Vipulanandan p-q to test and model the stress–strain behavior of oil well cement under compression and flexural stresses.
iv.    Constructing a new model to predict the modulus of elasticity of cement under varying strains.
v.     Conducting a new protocol test to determine the bonding strength between the cement and steel casing in the oil well.

## 2. Methodology

### 2.1. Cement Characterization

The cement composition was analyzed at 25 °C using XRD (Figure 3). The sample holder was packed to within 3 mm of its capacity with 2 g of cement powder. The XRD was evaluated at intervals of 0.02 from 10 to 90°. Scanning electron microscopy (SEM) was used to obtain information about the cement's morphology (Figure 4).

### 2.2. Sample Mixture

The samples were prepared following API standards with a w/c of 0.38. The results were verified by testing at least three samples for each condition.

### 2.3. Compressive Strength Test (ASTM C 39)

Compression tests were conducted on cement samples after 1, 7, and 28 days of curing using a hydraulic compression testing machine. The strain gauge used in this study had a 60 mm length and a resistance of 120 $\Omega$. A cylindrical mold measuring 50 mm in diameter and 100 mm in height. A stress–strain test was carried out at three different curing ages using a hydraulic compression testing system with a loading speed of 0.15 MPa/s. Specimens were instrumented with a strain gauge for lateral strain and an extensometer to measure axial strain to determine Poisson's ratio. The accuracy of the extensometer was checked with an axial strain gauge. Specimens were cast into appropriate molds for

the intended tests and then cured at room temperature in a relative humidity box with a humidity of at least 95%. The samples were demolded after 24 h and stored in the same conditions until testing.

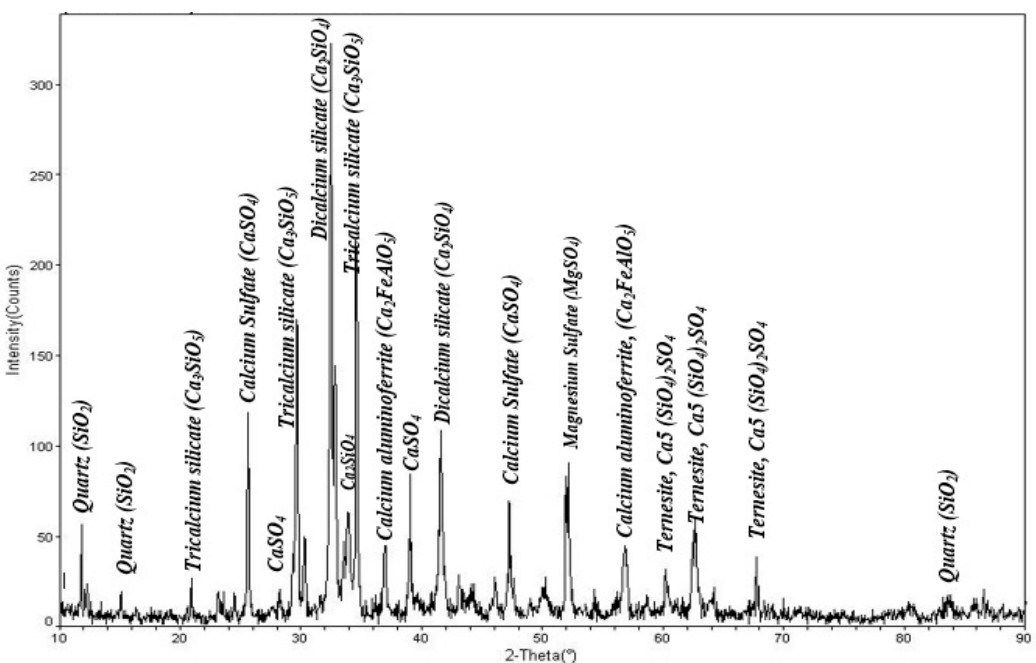

**Figure 3.** XRD for class H oil well cement.

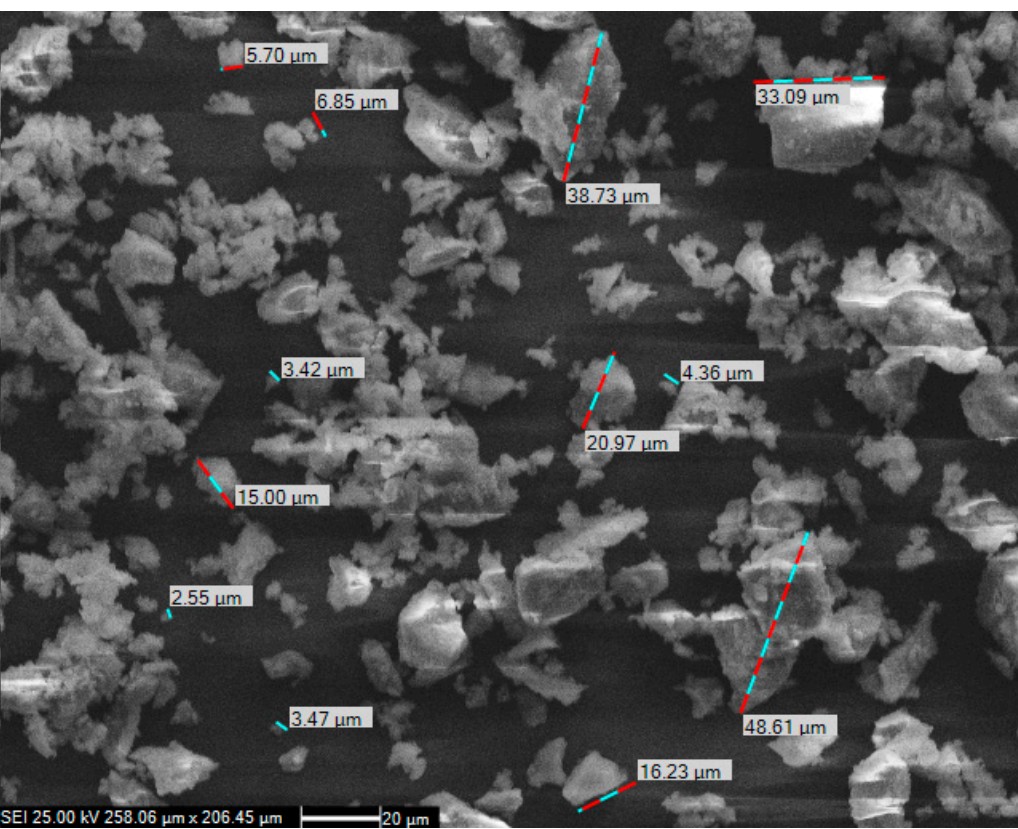

**Figure 4.** SEM for class H oil well cement.

### 2.4. Bonding between Steel Casing–Cement

Figure 5 depicts the experimental setup to evaluate the shear bonding between the steel casing and the cement. The steel casing was loaded up until it was flush with the cement. At a w/c of 0.38 and curing times of 1, 7, and 28 days, the applied load pushed the steel casing through the cement sheath. A portion of the steel enclosure was shifted aside (dia.: 45 mm; height: 74 mm). The load was applied to the steel casing until it reached the cement level (Figure 5b). The applied load pushed the casing through the cement sheath. Figure 5c shows the specimen at failure. The steel casing was moved to one side. After debonding from the cement sheath, the applied load acted against the friction between the steel–cement interface.

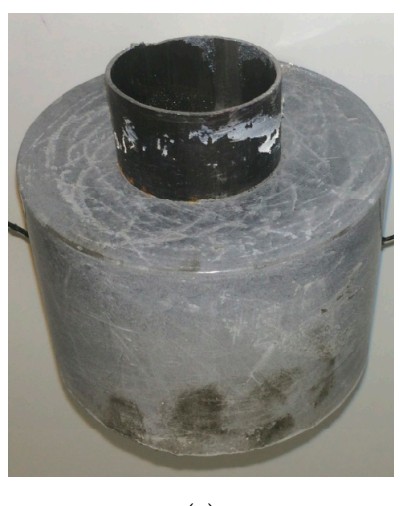 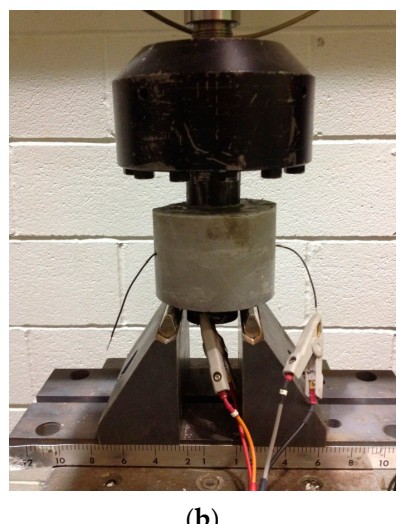 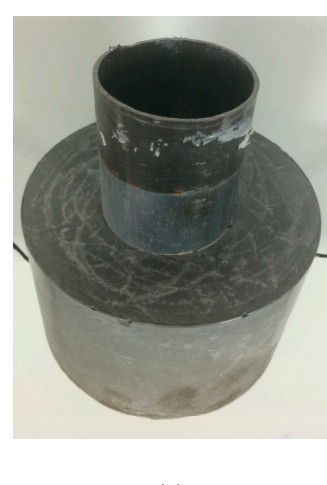

(**a**)  (**b**)  (**c**)

**Figure 5.** The specimen used to measure shear bonding between the steel casing and cement: (**a**) sample before the test, (**b**) specimen under the pull-out machine; (**c**) shear failure between casing and cement.

### 2.5. Flexural Strength Test

The ASTM C293/C293M-10 standard [20] called for a three-point bending test. The dimensions of the specimen beam were 280 × 76 × 76 mm. A Vernier caliper was used to measure the specimen's dimensions for stress analysis. Strain gauges were installed on the tension and compression fibers near the midspan. Equation (1) was utilized in the strength calculation:

$$\sigma_f = \frac{3PL}{2bd^2} \tag{1}$$

where $\sigma_f$ is the flexural stress, P is the applied load, L is the span length, b is the width of the specimen, and d is the depth of the sample. The strain was determined for each fiber by using strain gauges.

### 2.6. Fracture Toughness and CTOD

At least three samples were tested for each criterion, and the mean of those results is presented. A data-acquisition system was used to record the load (P), crack mouth opening displacement (CMOD), and load point deflection. The load cell's voltage output was used to determine the ideal value for the applied load. The strain was calculated from the two-wire DC resistance output of the strain gauge. Mode I fracture properties (Figure 2) were investigated using the beam specimen. As shown in Figure 6, a band saw was used to create a notch with a depth of 3 mm. A three-point loading setup was used to find the fracture toughness and crack tip opening displacement (CTOD) of different types of oil well cement. A clip on the CMOD gauge was used to measure the crack mouth opening displacement. Knife edges were used to clip the CMOD gauge to the pre-crack. At the same time, the resistance and pulse velocity was monitored to characterize the crack using

a nondestructive method. Midspan deflection also was monitored using LVDT. Using a three-point loading setup, the oil well cement was tested for fracture toughness and crack tip opening displacement (CTOD). The crack mouth opening displacement was measured using a clip-on CMOD gauge. The CMOD gauge was clipped to the pre-crack with the blades of knives. Figure 7 shows the overall experimental setup, which utilized a clip-on CMOD gauge to derive $K_I$ values. The schematic diagram of load vs. CMOD is presented in Figure 8.

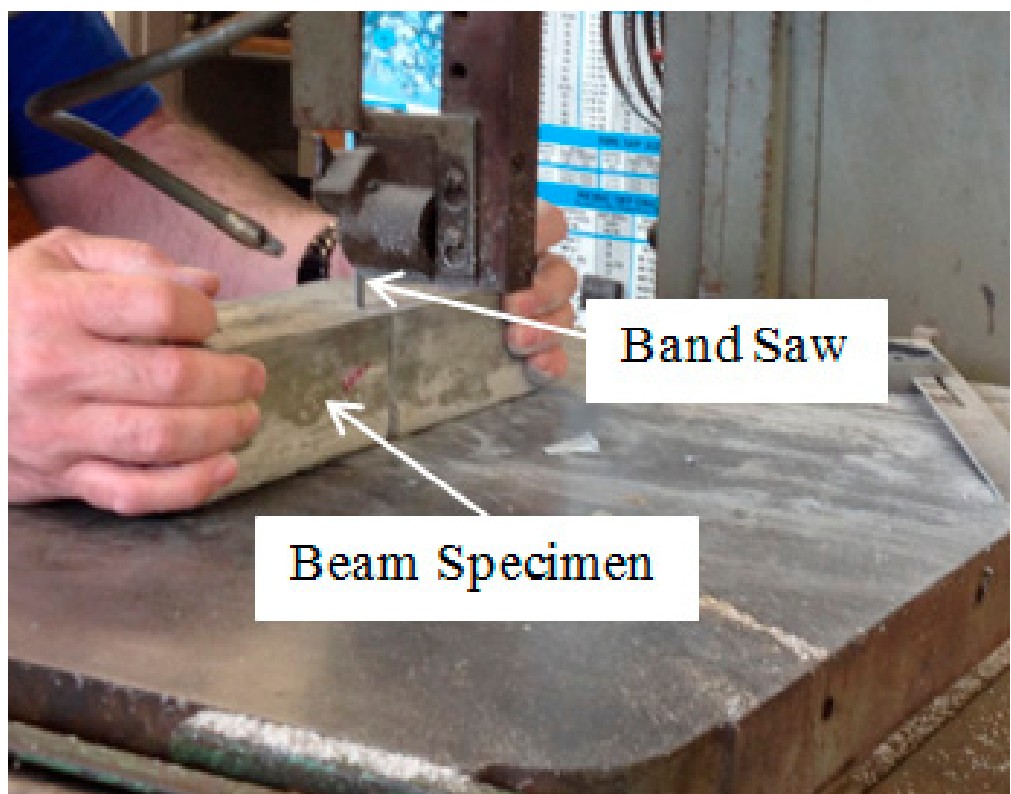

**Figure 6.** Making a notch in the specimen using hand saw.

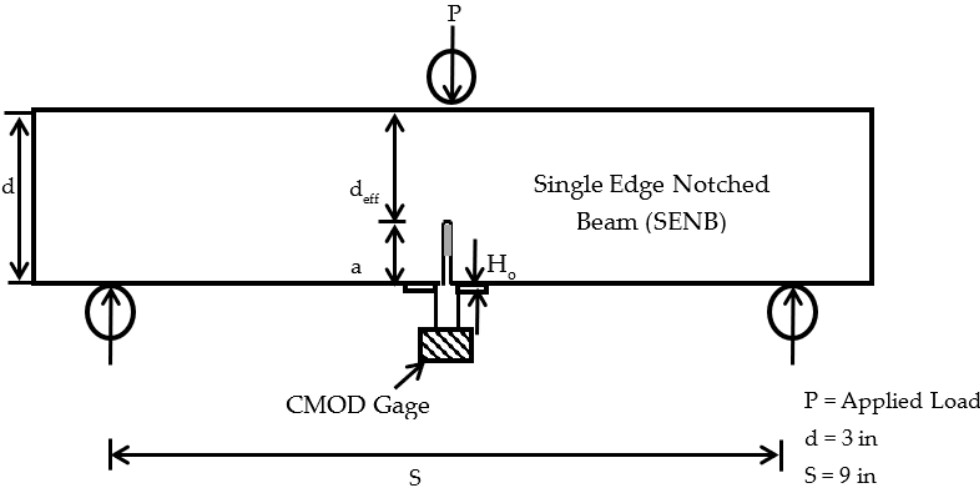

**Figure 7.** Schematic diagram of the experimental setup to find $K_I$.

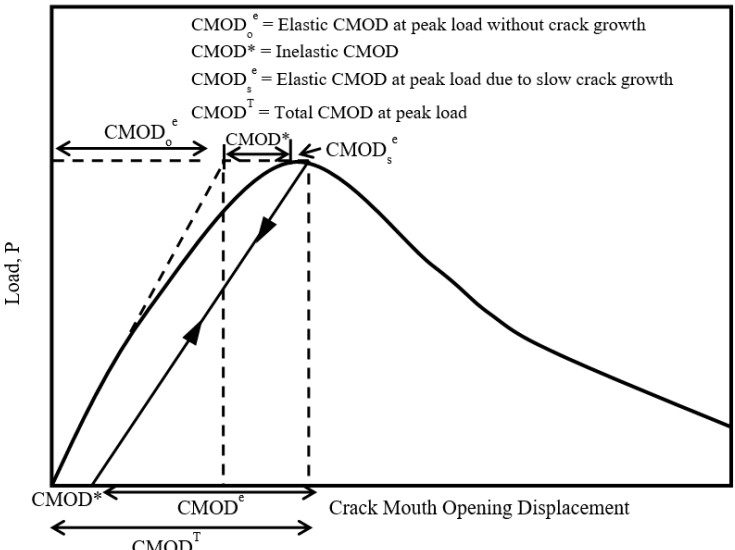

**Figure 8.** Schematic diagram of load versus CMOD with the components of $CMOD^T$.

For any geometries, Equation (2) provides the stress intensity factor $K_I$:

$$K_I = \sigma\sqrt{a}F(\alpha) \tag{2}$$

where $\sigma$ is the bending stress calculated as shown in Equation (1), $F(\alpha)$ is a finite width and loading geometry correction factor, and "a" is the height of the crack. For the three-point loading bending test, $F(\alpha)$ is given in Equation (3):

$$F(\alpha) = \frac{\left[1.99 - \alpha(1-\alpha)\left(2.15 - 3.93\alpha + 2.7\alpha^2\right)\right]}{\left[(1+2\alpha)(1-\alpha)^{\frac{3}{2}}\right]} \tag{3}$$

where $\alpha$ is a factor that depends on the crack length. Equation (4) defines $\alpha$ as:

$$\alpha = \frac{(a + H_0)}{(d + H_0)} \tag{4}$$

where $H_o$ is the clip gauge holder thickness, as shown in Figure 8. In Equation (4), 'a' must be calculated from a numerical iteration procedure using Equation (5):

$$a_e = a_0 \left[\frac{C_U}{C_0}\right]\left[\frac{V(\alpha_0)}{V(\alpha_e)}\right] \tag{5}$$

where initial compliance Co is given by Equation (6):

$$Co = \frac{CMOD}{P} \tag{6}$$

Unloading compliance $C_u$ was measured at about 95% of the peak load, and $V(\alpha)$ is given by Equation (7):

$$V(\alpha) = 0.76 - 2.28\alpha + 3.87\alpha^2 - 2.04\alpha^3 + \frac{0.66}{(1-\alpha)^2} \tag{7}$$

Variations in the load (P) versus the crack mouth opening displacement (CMOD) of the class H oil well cement in the experiment are shown in Figure 8 for the initial notch-to-depth ratios (a/d) of 0.3, 0.4, and 0.5 [18].

## 3. Results, Analysis, and Discussion

### 3.1. XRD and SEM

Major ingredients of the cement (Class H) included dicalcium silicate ($Ca_2SiO_4$), tricalcium silicate ($Ca_3SiO_5$), magnesium sulfate ($MgSO_4$), calcium aluminoferrite ($Ca_2FeAlO_5$), calcium sulfate ($CaSO_4$), and quartz ($SiO_2$) (Figure 3).

Figure 4 displays the results of the SEM analysis, which revealed that the diameter sizes of the cement particles ranged from 2.55 to 48.61 μm.

### 3.2. Vipulanandan p-q Model

The Vipulanandan p-q model was used to predict the stress–strain behavior of the soils, rocks, and piezoelectrical resistivity [21]. The ratio of the secant modulus to the initial tangential modulus is given by parameter q in Equation (8) (Table 1). The value of the model parameters was determined by minimizing the error in the stress–strain relationship. The model predicts a linear material up to peak stress when q equals 1. A smaller q value indicates a greater presence of nonlinear behavior. However, the post-peak behavior is controlled by the parameter p, while the pre-peak behavior is only moderately affected by it. Initially, the tangent and secant moduli were estimated to determine q. Then, the p values were adjusted to the lowest possible RMSE (Figure 9). Parameter q was defined as the ratio of the secant modulus at peak stress to the initial tangent modulus. Parameter p was obtained by minimizing the error in the predicted stress–strain relationship. The parameters p and q are influenced by the curing time of the cement. The shape of the stress–strain curve before and after the peak can be changed based on the p and q values, as shown in Figure 9.

$$\sigma = \frac{\frac{\varepsilon}{\varepsilon_f} * \sigma_f}{q + (1 - p - q)\frac{\varepsilon}{\varepsilon_f} + p\left(\frac{\varepsilon}{\varepsilon_f}\right)^{\frac{(p+q)}{p}}} \tag{8}$$

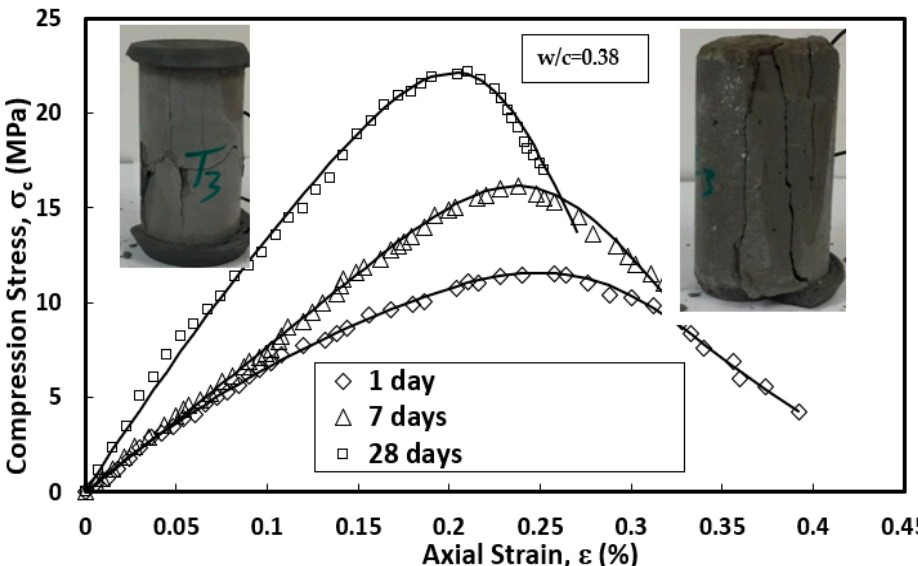

**Figure 9.** Compressive stress–strain behavior of the cement paste.

By differentiating Equation (8) and using the same p and q parameters for the stress-strain models in Equation (8), the modulus of elasticity of the cement paste versus strain

can be predicted using Equation (9) (s is the axial stress, $\sigma_f$ and $\varepsilon_c$ are the stress at failure and the corresponding strain, and p and q are model variables (Table 1)):

$$\frac{d\sigma}{d\varepsilon} = \frac{q*\left[1 - \left(\frac{\varepsilon}{\varepsilon_c}\right)^{\frac{(p+q)}{p}}\right] * \left(\frac{\sigma_c}{\varepsilon_c}\right)}{\left[\left[q*(1 - p - q) * \frac{\varepsilon}{\varepsilon_c}\right] + p*\left(\frac{\varepsilon}{\varepsilon_c}\right)^{\frac{(p+q)}{p}}\right]^2}$$ (9)

The model evaluations in Equations (10) and (11) are the coefficient of determination and the root mean square error, respectively:

$$R^2 = 1 - \frac{\sum_{i=1}^{n}\left(y_{experimental} - y_{predicted}\right)^2}{\sum_{i=1}^{n}\left(y_{experimental} - mean\right)^2}$$ (10)

$$RMSE = \left(\frac{\sum_{i=1}^{n}\left(y_{experimental} - y_{predicted}\right)^2}{N}\right)^{0.5}$$ (11)

### 3.3. Compressive Strength

The compressive stress–strain relationship for cement was predicted using the Vipulanandan p-q stress-strain model. The compressive strength ($\sigma_f$) of the cement after 1, 7, and 28 days of curing was 10.6, 15.8, and 18.3 MPa, respectively (Figure 9). The axial strain of the samples at failure varied between 0.21 and 0.3% (Table 1). The coefficient of determination ($R^2$) was 0.99 (Table 1). Differentiating Equation (8), the modulus of elasticity of the cement versus axial strain can be calculated using Equation (9), as shown in Figure 10. The compression modulus of elasticity of the cement at 1 and 28 days of curing times is summarized in Table 1. The same p and q values were obtained to predict the stress–strain behavior of the cement shown in Figure 9 and were used to predict the modulus of elasticity of the cement shown in Figure 10.

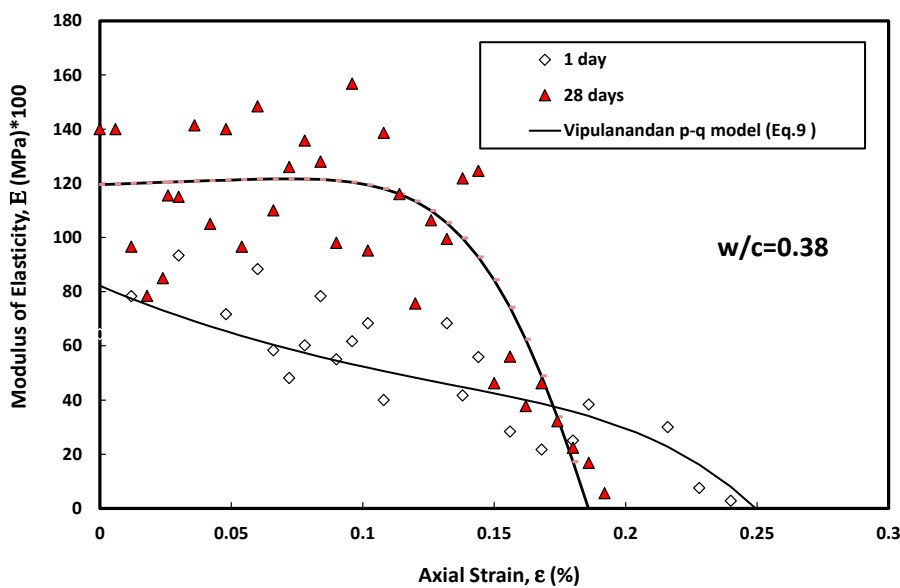

**Figure 10.** Relationship between axial strain and modulus of elasticity of the cement.

### 3.4. Fracture Properties of Cement

This study investigated the influence of utilizing a different a/d on the fracture behavior of cement at three different curing times and using the single-edge notch beam

in three-point bending determined parameters such as the s $K_I$ (mode 1), which has been frequently measured and widely reported for other mortars and concrete [13,22–24].

The experimental observations included the crack mouth opening displacement (CMOD), load versus load point displacement, and crack development due to applied loads.

### 3.4.1. Experimental Calculations

At different load levels (applied load), the following parameters could be calculated:

i.      Crack length (a);
ii.     Crack extension ($\Delta a$ = crack length − initial crack length = a − $a_i$);
iii.    Stress intensity factor ($K_I$);
iv.     Crack tip opening displacement (CTOD).

It should be mentioned that at peak load, the stress intensity factor and crack length were called the critical stress intensity factor ($K_I$) and effective crack length, respectively.

### 3.4.2. Calculation Procedures

For a three-point bend test specimen, by using the concept of the linear elastic fracture mechanic (LEFM), the crack length can be obtained from the relationship between the elastic crack mouth opening displacement ($CMOD^e$) and the corresponding crack length (a), as shown in Equation (12).

$$CMOD^e = 4\sigma\, a\, \frac{V(\propto)}{\acute{E}} \tag{12}$$

where:

$\sigma$ is the net stress, which is equal to 6 $M/bd^2$;
M is the applied pure bending moment;
$\propto$ is equal to $(a + H\circ)/(d + H\circ)$;
*a* is the crack length;
$\acute{E}$ for the plane, the stress is equal to E (Modulus);
$\acute{E}$ for the plane, the strain is equal to E$/(1 − v^2)$;
*v* is Poisson's ratio;
$V(\propto)$ can be calculated by using the following empirical formula:

$$V(\propto) = 0.8 − 1.7 \propto +2.4 \propto^2 +0.66 / (1 − \propto)^2 \tag{13}$$

The elastic crack mouth opening displacement could be determined at various loading levels if the crack grew slowly. Accordingly, using Equation (15), the corresponding crack length (*a*) could be evaluated. For a given measured load (P), the crack length (*a*) could be evaluated by applying a numerical iterative procedure such that the crack length (*a*) could be assumed in Equation (15) to obtain the CMOD calculated and then be compared with the CMOD measured. The procedure had to be repeated until the measured and calculated values of CMOD were in agreement.

Since the effective crack length $a_e$ was the sum of the initial notch plus an effective crack extension at the peak load, it can be mentioned that if the material behaved elastically up to the peak load without any crack extension, the relationship between the load and CMOD would be linear. However, suppose any displacement occurred due to unloading the specimen just before and immediately after the peak load. In that case, the displacement could be considered an inelastic displacement associated with the response of the notched beam. Studies have indicated that at peak load, the total CMOD ($CMOD^T$) is composed of the elastic displacement (no crack extension—CMOD $^e_\circ$), inelastic displacement ($CMOD^*$), and elastic displacement due to slow crack growth (CMOD $^e_s$).

The nonlinear displacement observed in the P-CMOD response could be attributed to creep and slow crack growth. To apply LEFM, the inelastic displacement ($CMOD^*$) had to be extracted from the total CMOD ($CMOD^T$) at peak load. The elastic ($CMOD^e$) at peak load was found by unloading the specimen immediately after peak load using the initial compliance ($C_i$) and the unloading compliance ($C_u$).

Where ($C_i$) is equal to (CMOD/P) and ($C_u$) is measured as about 95% of the peak load in Equation (15); by using the results, the effective crack length ($a_e$) could be determined using the following relationship:

$$a_e = a_i \left(\frac{C_u}{C_i}\right) \left[V_{(\alpha_i)} \, V_{(\alpha_e)}\right] \tag{14}$$

where $a_i$ is the initial crack length.

A numerical iterative procedure had to be used to estimate the effective crack length using Equation (14); thus, for a beam of cross-section (b x d) with crack length (*a*), the stress intensity factor ($K_I$) for the four-point bending could be calculated by applying the equation as follows:

$$K_I = \sigma\sqrt{a}\, y\left(\frac{a}{d}\right) \tag{15}$$

where $y(a/d)$ is a correction factor for finite width and loading geometry. For three-point bend tests:

$$y(\alpha) = \left[1.99 - \alpha\,(1-\alpha)\left(2.15 - 3.9\,\alpha + 2.7\,\alpha^2\right)\right] / \left[(1 + 2\,\alpha)(1-\alpha)^{\frac{3}{2}}\right] \tag{16}$$

Based on Equation (14) and by using the effective crack length ($a_e$), which was determined using Equation (15); instead of crack length (a) and peak load (P), the critical stress intensity factor ($K_{Ic}$) could be determined.

Figure 11 displays the load versus CMOD for each a/d at three curing times. The $CMOD^e$ was reduced by 39% when the a/d was increased from 0.3 to 0.5 (Figure 12a), and the value of the $CMOD^e$ increased by 11 to 36% when the curing time was increased from 1 to 28 days at the same a/d. Figure 11 depicts a linear relationship between the load and CMOD, assuming the material exhibited elastic behavior up to the peak load without any crack extension. This inelastic displacement (CMOD*) abruptly released the specimen after the peak load. The elastic displacement (no crack extension—$CMOD^e_o$), inelastic displacement (CMOD*), and elastic displacement due to slow crack growth at maximum load made up of the $CMOD^T$ ($CMOD^e_s$). The specimen was unloaded at 95% of the peak load to obtain the total elastic CMOD ($CMOD^e = CMOD^e_o + CMOD^e_s$). Figure 12a,b shows the relationships between curing time and $COMD^e$ and $K_I$ of oil well cement at three different a/b, respectively.

### 3.5. Flexural Stress–Strain

Equation (1) was used to model the flexural stress–strain behavior of the cement with a w/c of 0.38 up to 28 days of curing. When the cement was left to cure for 7 days, there was a 12% increase in flexural strength compared with 1 day of curing. After 28 days, there was a 15% increase in flexural strength compared with 7 days of curing, as shown in Figure 13. The Vipulanandan p-q model performed well in predicting the experimental data, with $R^2$ and RMSE values between 0.99 and 0.98 and 0.03 and 0.03 MPa, respectively (Table 1).

**Table 1.** Model parameters for stress–strain for oil well cement (w/c = 0.38).

| | Curing Age (Day) | σf (MPa) | εf (%) | Ei (MPa) | *p* | q | RMSE (MPa) | $R^2$ | Figure No. |
|---|---|---|---|---|---|---|---|---|---|
| Compressive | 1 | 10.6 ± 2 | 0.28 ± 0.01 | 9670 ± 21 | 0.034 ± 0.01 | 0.36 ± 0.02 | 0.128 | 0.99 | |
| | 7 | 15.8 ± 1 | 0.3 ± 0.02 | 12,000 ± 25 | 0.03 ± 0.02 | 0.36 ± 0.02 | 0.02 | 0.99 | Figure 9 |
| | 28 | 18.3 ± 1.5 | 0.21 ± 0.02 | 19,360 ± 20 | 0.160 ± 0.03 | 0.87 ± 0.04 | 0.11 | 0.99 | |
| Flexural | 1 | 1.36 ± 0.02 | 0.041 ± 0.002 | - | 71,199.4 ± 100 | 0.316 ± 0.02 | 0.03 | 0.99 | |
| | 7 | 2.32 ± 0.01 | 0.101 ± 0.001 | - | 71,214.6 ± 95 | 0.082 ± 0.01 | 0.05 | 0.99 | Figure 13 |
| | 28 | 3.0 ± 0.02 | 0.02 ± 0.002 | - | 71,138.8 ± 105 | 0.550 ± 0.03 | 0.10 | 0.99 | |

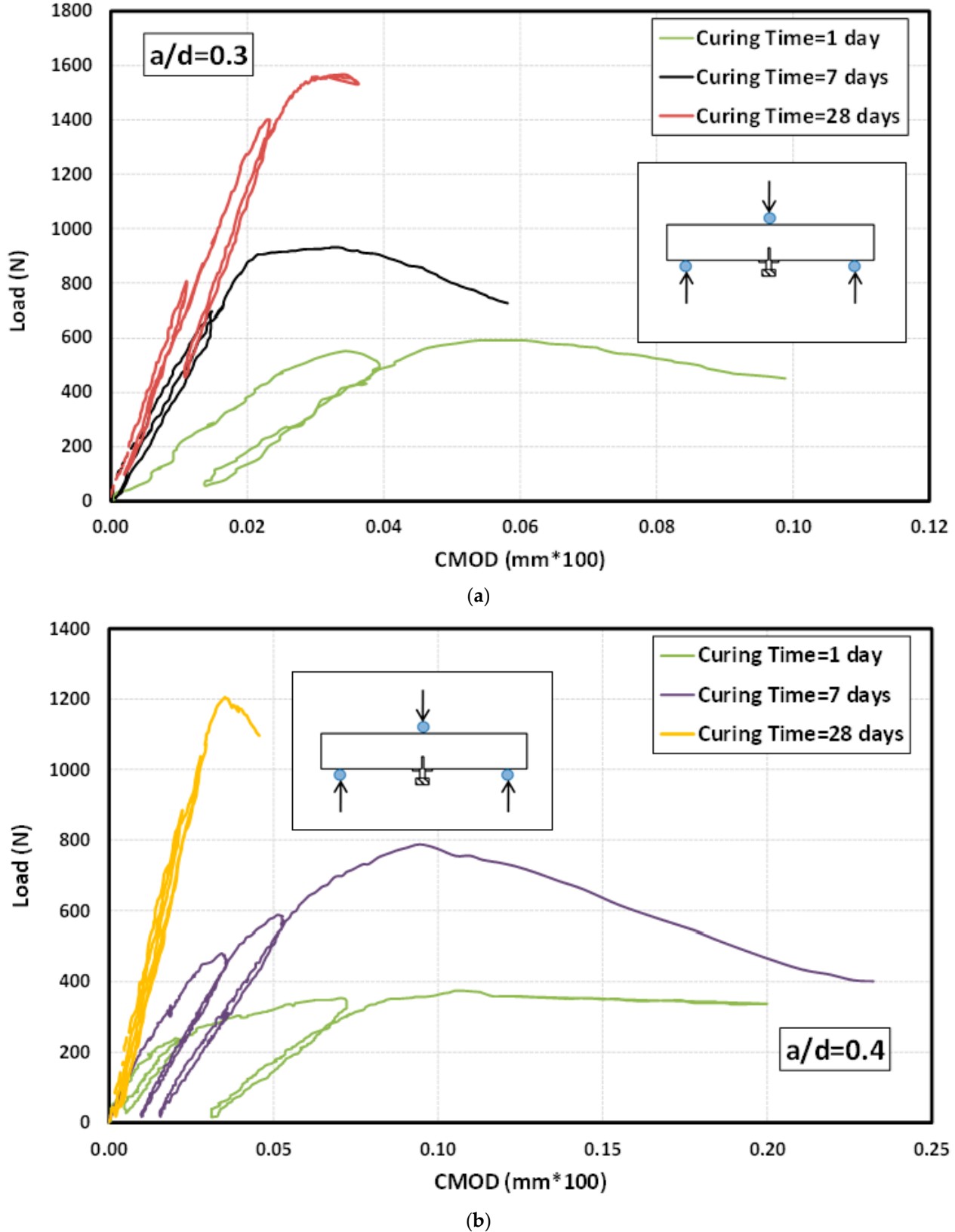

(**a**)

(**b**)

**Figure 11.** *Cont.*

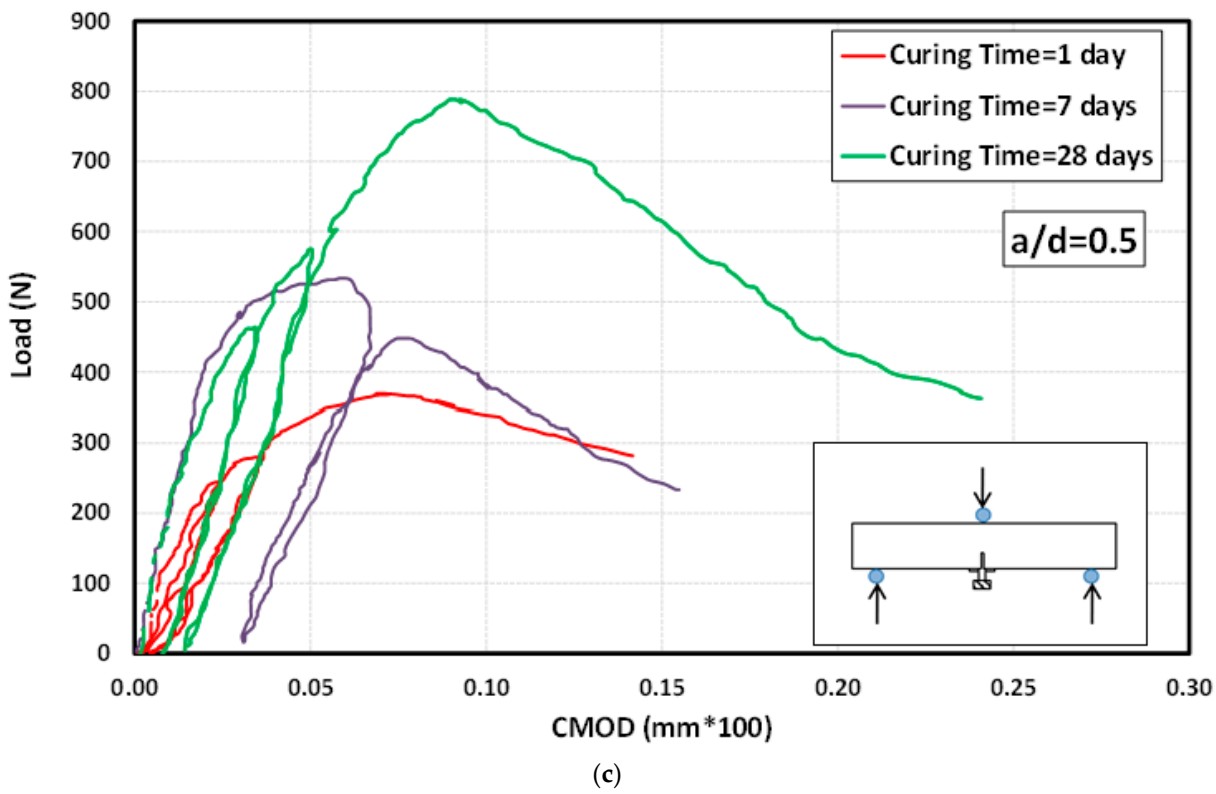

(**c**)

**Figure 11.** Relationship between axial strain and modulus of elasticity of cement: (**a**) a/b = 0.3 m; (**b**) a/b = 0.4; (**c**) a/b = 0.5.

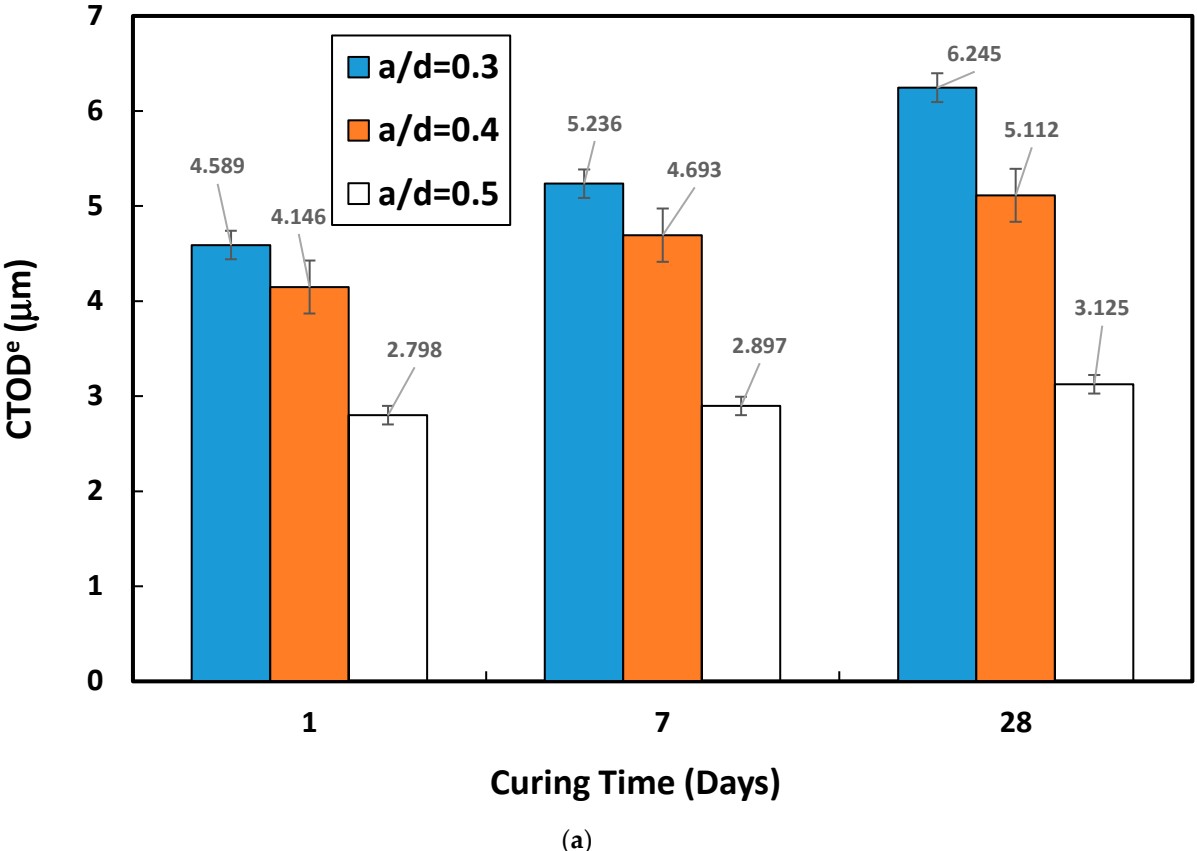

(**a**)

**Figure 12.** *Cont.*

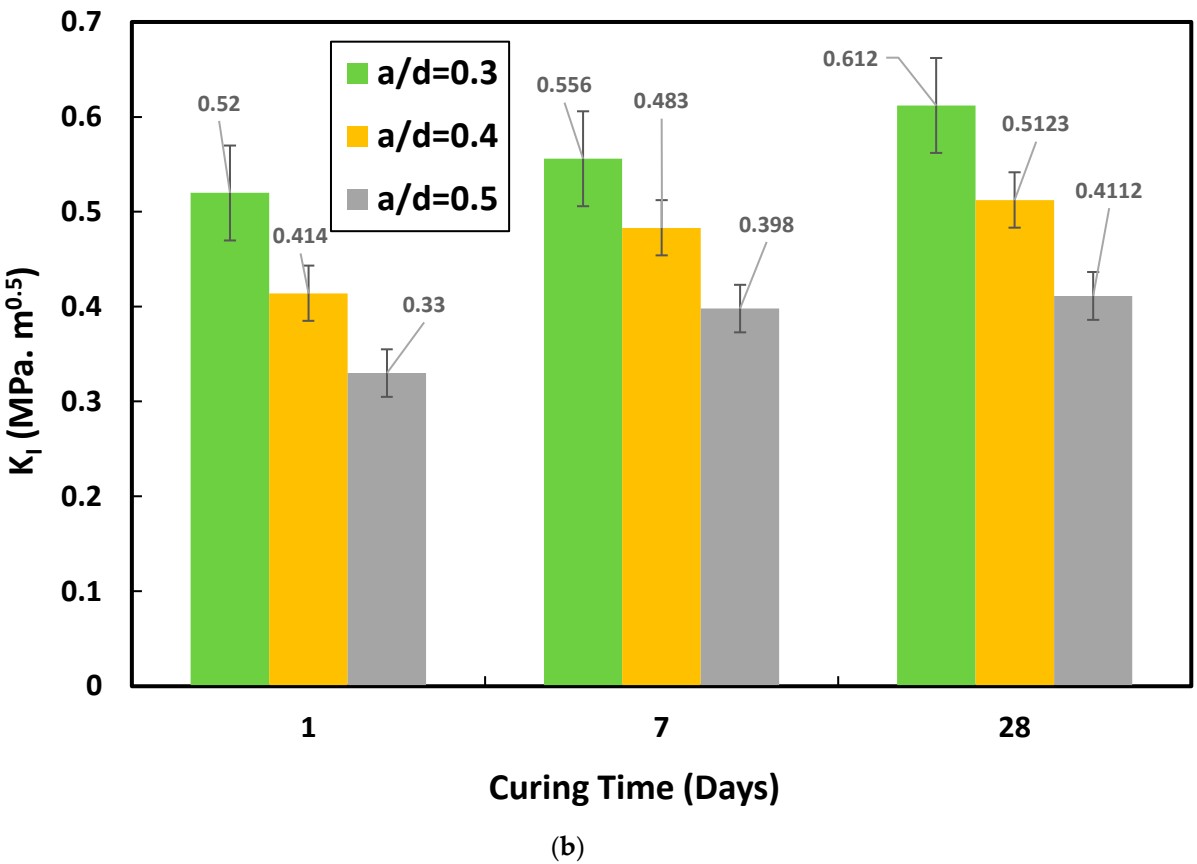

(**b**)

**Figure 12.** Variations in a/d ratio for oil well cement for (**a**) CTOD$^e$ and (**b**) K$_I$.

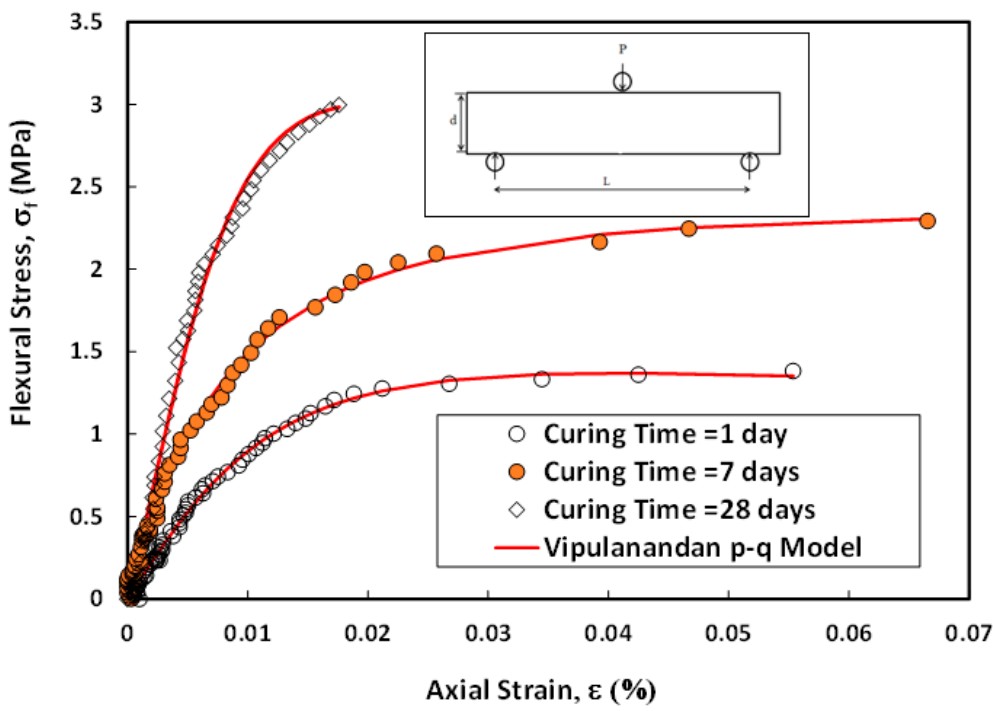

**Figure 13.** Flexural stress–strain for the oil well cement at three different curing times.

The K$_I$ increased as the notch-to-depth ratio grew (Figure 14b). In this study, the K$_I$ ranged from 0.33 to 0.612 MPa.m$^{0.5}$. The results of this study were consistent with previous reports that the K$_{IC}$ of cement pastes, cement mortars, and cement concretes can range

from 0.1 to 1.7 MPa.m$^{0.5}$. It was hypothesized that the wide variety of K$_I$ values was due to the variations in the material that caused scatter in experimental results and the limitations in the applicability of fracture mechanics concepts. The K$_I$ of the cement paste decreased with the ratio of a/d; also, the K$_I$ increased from 0.33 at 1 day of curing to 0.398 and 0.4112 MPa.m$^{0.5}$ when the curing time was increased to 7 and 28 days, respectively.

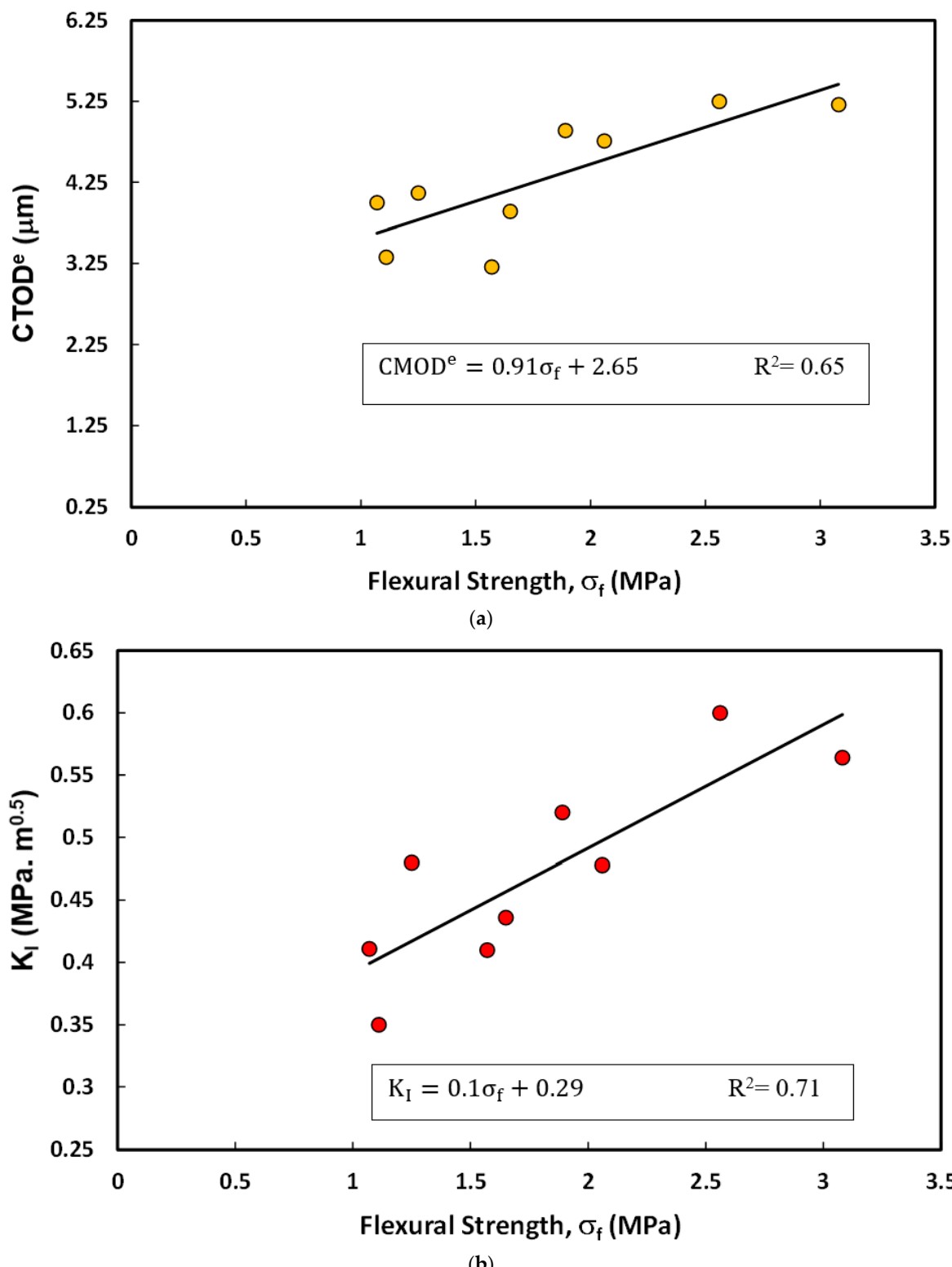

**Figure 14.** Correlations between flexural strength and (**a**) CTODe and (**b**) K$_I$.

The following correlations were developed between the CTOD$^e$ and K$_I$ with the flexural strength ($\sigma_f$) of the cement paste (Figure 14):

$$\text{CMOD}^e = 0.91\sigma_f + 2.65 \quad \text{R}^2 = 0.65 \tag{17}$$

$$\text{K}_I = 0.1\sigma_f + 0.29 \qquad \text{R}^2 = 0.71 \tag{18}$$

*3.6. Bonding Strength*

It is essential to consider not only short-term well integrity factors such as cement quality and pumpability at the time of cementing operation but also the long-term integrity of the cement sheath, including the cement/casing bonding throughout the well's life and developing a high bonding strength between the cement and steel. The casing between the cement and the formation on the other side is an essential property determining the life of oil and gas wells. This new protocol test measured the bonding strength between the steel tube and the oil well cement at three curing times (1, 7, and 28 d). The respective bonding strengths were 0.35, 1.39, and 3.59 MPa for the three curing times (Figure 15). The steel casing was moved to one side. After debonding from the cement sheath, the applied load acted against the friction between the steel–cement interface. Mechanical properties were tested by conducting compression, indirect tension, flexural, fracture toughness, and steel casing–cement bonding tests.

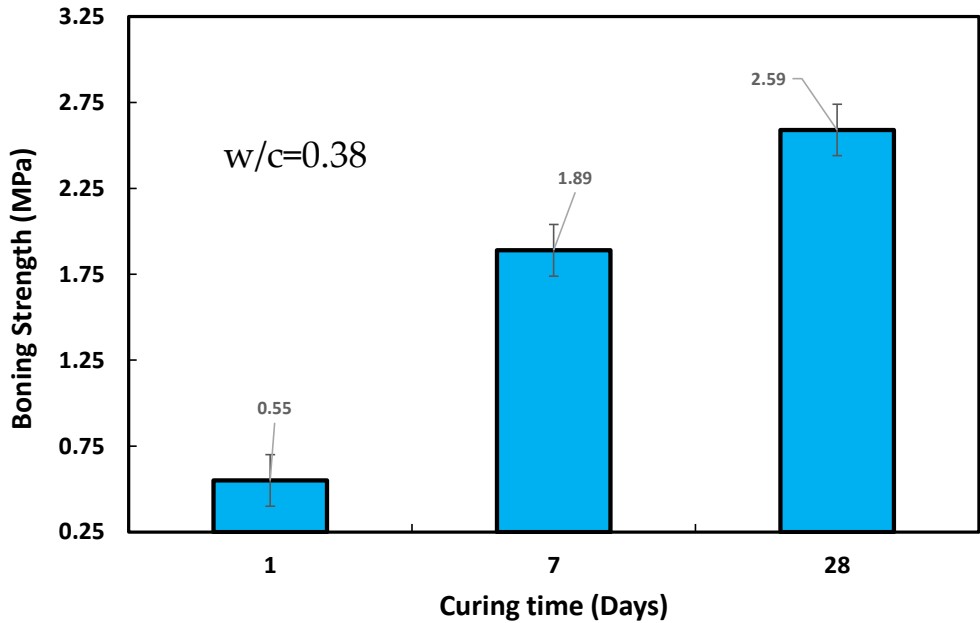

**Figure 15.** Relationship between bonding strength and curing time.

**4. Conclusions**

This study aimed to determine the mechanical properties of an oil well cement paste and its fracture behavior. The following conclusions were obtained from this study.

1. The compressive and flexural strengths for the oil well cement at a w/c of 0.38 increased from 10.6 to 18.3 MPa and from 1.36 to 3 MPa, respectively, when the curing time was increased from 1 to 28 days of curing.
2. Studies of fractures showed that the K$_I$ values ranged from 0.3 to 0.6 MPa.m, and the CMOD values ranged from 2 to 6 m.
3. Increasing the ratio of the crack mouth depth and the thickness of the beam (a/d) from 0.3 to 0.5 decreased the CMOD$^e$ and the K$_I$ by 39–50% and 33–37% based on curing time, respectively. Based on the flexural strength of the oil well cement, the CMOD$^e$ and K$_I$ could be estimated from the developed linear relationship.

4. Based on the $R^2$ and RMSE, the Vipulanandan p-q model accurately predicted the tested oil well cement's compressive and flexural stress–strain behaviors.
5. More studies on the effect of the w/c on the bonding, compression, and flexural strengths of oil well cement should be conducted to justify the behavior of the oil well cement under different environmental conditions in an oil well.

**Author Contributions:** C.V. and A.S.M. collected the data, conducted planning, and wrote the manuscript; P.R. was responsible for the results and analysis; A.S.M. and C.V. were responsible for the conclusions and editing of the manuscript. All authors have read and agreed to the published version of the manuscript.

**Funding:** This research received no external funding.

**Informed Consent Statement:** Informed consent was obtained from all subjects involved in the study.

**Data Availability Statement:** The data supporting the conclusions of this article are included in the article.

**Acknowledgments:** Center for Innovative Grouting Materials and Technology (CIGMAT), and Texas Hurricane Center for Innovative Technology in the University of Houston, TX-USA, CIGMAT Laboratory supported this work.

**Conflicts of Interest:** The authors declare no conflict of interest.

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
