# Peer review of "Experimental Study and Modeling of the Fracture Behavior, Mechanical Properties, and Bonding Strength of Oil Well Cement"

_sustainability, doi:10.3390/su15129566_

Round 1
Reviewer 1 Report
In this paper, fracture mechanics, compressive, flexural, and bonding strengths characterizations of cement paste were investigated. There are a lot of typographical errors. I recommend the publication of this manuscript after major revision.
1) The section of introduction is too long.
2) In the section of introduction, the novelty and motivation of the work needs to be described further.
3) “1.1. Research Significants”?
4) Page 4, line 157 “degrees Celsius”?
5) Please provide the information of testing equipments.
6) The serial number of figures is out-of-order.
7) What does “Flexural Properties” mean?
8) Are Equation 1 and 11 same?
9) Page 6, line 247 what does “NanoAl2O3” refer to?
10) Please add further discussion about bonding strength.
Author Response
Reviewer #1:
In this paper, fracture mechanics, compressive, flexural, and bonding strengths characterizations of cement paste were investigated. There are a lot of typographical errors. I recommend the publication of this manuscript after major revision.
1) The section of introduction is too long.
Response: Thank you very much for the observations made by reviewer #1. The introduction has been shortened based on your suggestion.
2) In the section of introduction, the novelty and motivation of the work needs to be described further.
Response: Based on your comment, the novelty and importance of studying crack behavior under cycling load in the cement and concrete have been added to the paper.
3) “1.1. Research Significants”?
Response: fixed, as you suggested.
4) Page 4, line 157 “degrees Celsius”?
Response: fixed, as you suggested.
5) Please provide the information of testing equipment.
Response: more information on the test procedure and the devices are added to the paper and highlighted.
6) The serial number of figures is out-of-order.
Response: fixed, and thanks.
7) What does “Flexural Properties” mean?
Response: it should be flexural strength instead of flexural properties. Fixed and thanks
8) Are Equation 1 and 11 same?
Response: Yes, they are the same but presented in two ways, Eq.1 presents the flexural stress at failure, and Eq. 11 represents the ultimate flexural stress.
9) Page 6, line 247 what does “NanoAl2O3” refer to?
Response: it refers to Al2O3 nanoparticles. The section has been modified.
10) Please add further discussion about bonding strength.
Response: more information on the bonding strength was added to the paper and highlighted.

Reviewer 2 Report
The major fault of the paper is it did not explain the distinguishing feature of the Vipulanandan p-q model. Can this methodology be applied only to oil well cement? What about Portland cement? The lengthy introduction is filled with unnecessary information, but it does not describe the purpose of the study. I believe these points should be courteously answered before the manuscript is accepted. Please consider the advice/opinions/questions given below for further improvement.
1. Page 14 is unclear, because of interceded Figures. Please read carefully before submitting the manuscript.
2. 3.1: What is the purpose of measuring XRD?
3. Line 249: “Figure 5 displays the results of the SEM analysis,” There is no Figure 5 in this manuscript.
4. Line 490: The Figure caption is wrong
5. 2.1: Figure numbers are wrong.
6. Equation 11 is enough to calculate flexural strength. Equations 8, 9, and 10 are not necessary to show in this paper. They are basics that are suitable for undergraduate engineering students.
7. Line 392: What is the “compressive flexural strengths”?
8. What is the connection of Reference 17 to this manuscript?
9. There are careless mistakes. Please correct them.
Examples:
Line 244: (MgSO4)
Line 245: (CaSO4)
Line 411: This work had no finding.
Please write in correct English. Line 141: Research Significants→Significance
Author Response
Reviewer #2:
The major fault of the paper is it did not explain the distinguishing feature of the Vipulanandan p-q model. Can this methodology be applied only to oil well cement? What about Portland cement? The lengthy introduction is filled with unnecessary information, but it does not describe the purpose of the study. I believe these points should be courteously answered before the manuscript is accepted. Please consider the advice/opinions/questions given below for further improvement.
Response: Thank you very much for your comments. The model has been used in different research areas, such as predicting the compressive strength of rock, Portland cement, drilling muds, UHPFRC, soils, and piezoelectrical resistivity in our research group. The followings are the articles published in high-level journals for using the Vipulanandan p-q model:
https://doi.org/10.4043/28880-MS
https://doi.org/10.1016/j.ejpe.2018.07.001
https://doi.org/10.1080/19386362.2018.1468663
https://doi.org/10.1016/j.rinma.2019.100011
https://doi.org/10.4043/28947-MS
https://doi.org/10.1007/s10706-018-0633-5
doi.org/10.1515/eng-2019-0055
https://doi.org/10.1016/j.istruc.2021.01.063
- Page 14 is unclear, because of interceded Figures. Please read carefully before submitting the manuscript.
Response: The figures have been reorganized and fixed.
- 3.1: What is the purpose of measuring XRD?
Response: To determine the crystallographic structure of cement, identify crystalline phases and orientation and determine the structural properties of cement.
- Line 249: “Figure 5 displays the results of the SEM analysis,” There is no Figure 5 in this manuscript.
- Line 490: The Figure caption is wrong, 5. 2.1: Figure numbers are wrong.
Response: Based on your comment, the figure numbers have been checked, corrected, and reorganized.
- Equation 11 is enough to calculate flexural strength. Equations 8, 9, and 10 are not necessary to show in this paper. They are basics that are suitable for undergraduate engineering students.
Response: Removed based on your suggestion
- Line 392: What is the “compressive flexural strengths”?
Response: it is compressive and flexural strengths. Fixed, and thanks.
- What is the connection of Reference 17 to this manuscript?
Response: Reference 17 has been replaced with “Crack evolution characteristics of cemented-gangue–fly-ash backfill with different proportions of fly ash and cement”
- There are careless mistakes. Please correct them.
Examples:
Line 244: (MgSO4), Line 245: (CaSO4), Line 411: This work had no finding.
Response: fixed, and thanks.
Comments on the Quality of English Language
Please write in correct English. Line 141: Research Significants→Significance
Response: The texts, including the abstract, tables, and figures, have been edited and organized regarding conversions, spelling, nouns, variety, word order, punctuations, prepositions, and fluency. The paper has been modified to satisfy the scientific journal requirements.

Reviewer 3 Report
After reading the manuscript, one major question arose to the reviewer’s mind. How good is this cement for oil well purposes? Is it the best? Comparison with regular cements and other oil well cements performance is missing.
Despite of cyclic loading being mentioned in the introduction, it is not clear in the text if it was really carried out.
The look of the figure’s section is unspeakable. It is hard to believe that someone can submit a manuscript in these conditions.
Although the relevance of the author’s research in the field of modelling the stress-strain behaviour is acknowledged, self-citation on 13 of 23 references is maybe a bit too much.
Several issues that need to be addressed may be found in the attached file.

Author Response
Reviewer #3:
After reading the manuscript, one major question arose to the reviewer’s mind. How good is this cement for oil well purposes? Is it the best? Comparison with regular cements and other oil well cements performance is missing.
Response: Oil-well cementing is the process of pumping cement slurry down the annulus between the casing and wellbore. This is known as primary cementing. Along with supporting the casing in the wellbore, the cement is designed to isolate zones, keeping each penetrated zone and its fluids from communicating with other zones. When designing a cement job, it is critical to consider the wellbore and its properties to keep the zones isolated.
Despite of cyclic loading being mentioned in the introduction, it is not clear in the text if it was really carried out.
The look of the figure’s section is unspeakable. It is hard to believe that someone can submit a manuscript in these conditions.
Although the relevance of the author’s research in the field of modelling the stress-strain behaviour is acknowledged, self-citation on 13 of 23 references is maybe a bit too much.
Several issues that need to be addressed may be found in the attached file.
Response: Thanks for your comments which have improved the quilty of the paper with much more information. The table and figure problem occurred when the system converted the manuscript from the normal style to the journal template. All the figure captions have been misted and become hard to follow. After responding to your comments, the paper has reached the high-level journal requirements.
The comments in a pdf file have also responded.

Reviewer 4 Report
This paper shows an investigation on the fracture mechanics, compressive, flexural, and bonding strengths characterizations of cement paste. Generally, this paper is well organized and can be considered after revision. The following comments should be addressed.
(1) The related investigations on cement paste have been reported by a mass of researchers in previous studies. The title “cement paste” can be replaced with “oil well cement paste”, or some other keywords can be provided in title, making it more different.
(2) Most of the references are outdated, and the references in recent 3 years should be added. Besides, there are two references named [1], which should be checked.
(3) The author should give a general review on the performance of cement-based materials. The following references may be helpful: https://doi.org/10.1016/j.cemconcomp.2022.104629 and https://doi.org/10.1016/j.conbuildmat.2023.131328
(4) The author should clear the topic related to the sustainability, which meets the requirement of this journal.
(5) The correlation coefficient R2 in Fig. 13 is not clear. Please check it.
Moderate editing of English language required
Author Response
Reviewer #4:
This paper shows an investigation on the fracture mechanics, compressive, flexural, and bonding strengths characterizations of cement paste. Generally, this paper is well organized and can be considered after revision. The following comments should be addressed.
(1) The related investigations on cement paste have been reported by a mass of researchers in previous studies. The title “cement paste” can be replaced with “oil well cement paste”, or some other keywords can be provided in title, making it more different.
Response: Thanks for the comment. The title has been modified based on your comment.
(2) Most of the references are outdated, and the references in recent 3 years should be added. Besides, there are two references named [1], which should be checked.
Response: Fixed and thanks.
(3) The author should give a general review on the performance of cement-based materials. The following references may be helpful: https://doi.org/10.1016/j.cemconcomp.2022.104629 and https://doi.org/10.1016/j.conbuildmat.2023.131328
Response: Thanks for the reviewer's useful references, which have improved the paper with much more information. The papers have been added and cited in the paper.
(4) The author should clear the topic related to the sustainability, which meets the requirement of this journal.
Response: Oil well cement plays an important part in well integrity. It is an essential barrier to keeping well in good working conditions for a long time. However, the cement can be affected during the production life of the wellbore. This could include problems that may break this important barrier which may lead to the loss of the sustainability of the well. Oil well cement cracks are one of the most common failures in oil and gas wells. Cracks can reduce cement strength, resulting in a loss of zonal isolation and fluid leak.
Recently this paper related to the crack in the cement was published in this journal (https://www.mdpi.com/2071-1050/15/1/577)
(5) The correlation coefficient R2 in Fig. 13 is not clear. Please check it.
Response: Fixed and Thanks
Comments on the Quality of English Language, Moderate editing of English language required
Response: The texts, including the abstract, tables, and figures, have been edited and organized in conversions, spelling, nouns, variety, word order, punctuation, preposition, and fluency. The paper has been modified to satisfy the scientific journal requirements. More scientific information and details were added to the paper to satisfy the reader.

Round 2
Reviewer 1 Report
The paper has been revised according to the comments.
Author Response
Reviewer Responses
Manuscript Number: sustainability-2422001R2
Dear Editor:
The paper has improved with much more supporting information thanks to the editor and reviewer's comments. The article has been modified based on the review comments. All the reviewer comments (point by point) have been considered during the first and second rounds of reversion, and the modifications are highlighted in the paper.
Reviewer #1:
The paper has been revised according to the comments.
Response: Thank you very much for your efforts and time during the revision.
Reviewer 2 Report
Again, there are many mistakes, though some places are marked in yellow.
Examples:
Line 179: 1.1. Research Significants
Section 2.1: Figure numbers are still wrong (XRD- Diffraction (Fig.4)).
Line 86: The following are some of the conditions that could causes damage to the cement sheath (causes→cause)
Line 264: “a” is the high of crack (high→height)
English is grammatically wrong in some places.
Example:
Line 86: The following are some of the conditions that could causes damage to the cement sheath (causes→cause)
Author Response
Reviewer Responses
Manuscript Number: sustainability-2422001R2
Dear Editor:
The paper has improved with much more supporting information thanks to the editor and reviewer's comments. The article has been modified based on the review comments. All the reviewer comments (point by point) have been considered during the first and second rounds of reversion, and the modifications are highlighted in the paper.
Reviewer #2:
Again, there are many mistakes, though some places are marked in yellow.
Examples:
Line 179: 1.1. Research Significants
Response: Sorry for the Inadvertent error. Fixed and thanks
Section 2.1: Figure numbers are still wrong (XRD- Diffraction (Fig.4)).
Response: Fixed and thanks.
Line 86: The following are some of the conditions that could causes damage to the cement sheath (causes→cause)
Response: Fixed, and thank you very much.
Line 264: “a” is the high of crack (high→height)
Response: Fixed, and thank you very much.
Comments on the Quality of English Language, English is grammatically wrong in some places.
Example:
Line 86: The following are some of the conditions that could causes damage to the cement sheath (causes→cause)
Response: The texts, including the abstract, tables, and figures, have been edited and organized regarding conversions, spelling, nouns, variety, word order, punctuations, prepositions, and fluency using editing software. The paper has been modified to satisfy the scientific journal requirements.
Reviewer 3 Report
The authors did not show the ability to improve the manuscript that it becomes acceptable for the journal standards. Therefore, the recommendation is to reject the manuscript.
The authors should have provided a specific reply to each reviewer’s comment.
The response to the first comment is a description of the cementing process instead of a performance comparison with other oil weel and ordinary cements. This compromises the significance of the research.
No response to the second comment.
No response to the fourth comment. Nevertheless, it is acknowledged that the self-citation ratio has lowered to 46%.
Ln 84-89 It is still not clear whether the described conditions are causes or consequences.
Ln 97-100 It is still not clear whether the microcracks are formed in fresh, hardened or both pastes.
Ln 115-120 It is hard to believe that “cracking of the cement sheath” and "plastic deformations” can be considered “interfaces”.
Ln 179 The title is still wrong.
Ln 207-208 The sentence does not make sense.
Ln 211-212 the information is redundant and not accurate. One inch is not exactly 25 mm.
Ln 248 Now it is still more confusing. Is it 2.5? Is it 3.0? The doubt if there is one or three notch-to-depth ratios remains. Or was the notch constant and the specimen depth variable?
Ln 264 “high of crack”?
Ln 280-281 Was not this checked before validating the submission?
Ln 297-306 The authors still not justify the preference for estimating one parameter and then minimizing the error for the second instead of using the Levenberg-Marquadt algorithm
Ln 363-364 is not descriptions do not seem correct
There are no significant comments on the quality of English.
Author Response
Reviewer Responses
Manuscript Number: sustainability-2422001R2
Dear Editor:
The paper has improved with much more supporting information thanks to the editor and reviewer's comments. The article has been modified based on the review comments. All the reviewer comments (point by point) have been considered during the first and second rounds of reversion, and the modifications are highlighted in the paper.
Reviewer #3:
The authors did not show the ability to improve the manuscript that it becomes acceptable for the journal standards. Therefore, the recommendation is to reject the manuscript.
Response: The authors believe that the paper reached a high-level journal after responding to the reviewers commnets during the first and second revisions. Thank you very much for your time.
The authors should have provided a specific reply to each reviewer’s comment. The response to the first comment is a description of the cementing process instead of a performance comparison with other oil weel and ordinary cements. This compromises the significance of the research.
Comment:
After reading the manuscript, one major question arose to the reviewer’s mind. How good is this cement for oil well purposes? Is it the best? Comparison with regular cements and other oil well cements performance is missing.
Response from the authors: Oil well cement is specialized for sealing oil or gas well drilling operations. Together with additives, Portland cement or mixed cement are used to make it. It must be slow-setting and able to survive the extreme heat and pressures seen in these deep wells. By filling the area between the rocks and the steel casing, you may isolate various well zones, stop leaks, stop corrosion, and prevent leaks. To satisfy particular criteria, oil well cement is produced using a variety of grades and methods.The slurry mentioned above has to be injected into place and set at a significant depth. Here, the temperature may reach 175 °C, and the pressure could reach 1300 kg/cm2. In these circumstances, cement slurry must maintain a sufficient flow for many hours in order to be pumped. As a result, the water-to-cement ratio is maintained.
Additionally, the cement setting has to be slowed down. Reducing the C3A concentration and fineness of cement will speed up the setting process. The slurry must, however, swiftly set and harden after being injected into position. Because of sulphur fumes or water that contains dissolved salts, the cement slurry must be able to withstand corrosive conditions. Oil well cement is the kind of cement that satisfies all of these conditions and may be used in an oil well. For this use, modified regular Portland cement is produced. In order to meet the unique needs of oil wells, oil well cement is a modified version of Portland cement that fills the area between the steel casing of the well and the rocks.
No response to the second comment.
Response from the authors (Comment) Despite cyclic loading being mentioned in the introduction, it is unclear in the text if it was really carried out.
Sorry for the Inadvertent error. Failure of the cement sheath seal will result in problems, including fluid leakage and persistent casing pressure. A somewhat risky working environment for the downhole cement sheath is created by cyclic loading, which results in cumulative plastic strain and strength deterioration of cement stone. The paragraph in the text has been modified based on your comment.
No response to the fourth comment. Nevertheless, it is acknowledged that the self-citation ratio has lowered to 46%.
Response from the authors: Based on your recommendation, the number of self-cations was reduced to about 16%.
Ln 84-89 It is still not clear whether the described conditions are causes or consequences.
Response: The paraghraph has been modified to the following based on your commnet” “The safety and effectiveness of the CO2 injection procedure for geologic carbon storage depends on the integrity of the cement, which offers zonal isolation and mechanical support. This research focuses on radial cracking in cement after CO2 injection and interfacial debonding at wellbore contacts. It applies the energy release rate (ERR) definition to describe how fractures spread. The suggested model, which uses the finite element approach, calculates the ERRs of both kinds of fractures using realistic wellbore layouts and injection settings. Additional parametric research reveals how the fracture geometry, cement's mechanical and thermal characteristics, and crack size affect crack propagation. According to simulation findings, with normal cement characteristics, interfacial and radial fracture ERRs would be more than 100 J/m2. The Young's modulus, Poisson's ratio, and thermal conductivity of the cement are the next most significant influences on the ERR. Another crucial factor in regulating fracture propagation is the cracks' starting sizes and locations. Furthermore, the fracture propagation at the interfaces would be accelerated by non-uniform in situ loads. These important results might contribute to the improvement of cement sheath design and guarantee the long-term integrity of wells used for geological carbon storage.
Ln 97-100 It is still not clear whether the microcracks are formed in fresh, hardened or both pastes.
Ln 115-120 It is hard to believe that “cracking of the cement sheath” and "plastic deformations” can be considered “interfaces”.
Ln 179 The title is still wrong.
Response:The new proposed title is “Experimental study and modeling of the fracture behavior, mechanical properties, and bonding strength of oil well cement”.
Ln 207-208 The sentence does not make sense.
Response: The sentence has been modified and clarified.
Ln 211-212 the information is redundant and not accurate. One inch is not exactly 25 mm.
Response: Fixed and thanks.
Ln 248 Now it is still more confusing. Is it 2.5? Is it 3.0? The doubt if there is one or three notch-to-depth ratios remains. Or was the notch constant and the specimen depth variable?
Response: The height of the beam was constant (d constant) only the height of the crack was different (a was different); in this study, three a/d were considered.
Ln 264 “high of crack”?
Response: Fixed and thanks.
Ln 280-281 Was not this checked before validating the submission?
Response: Fixed, highlighted in the text and thanks
Ln 297-306 The authors still not justify the preference for estimating one parameter and then minimizing the error for the second instead of using the Levenberg-Marquadt algorithm
Response:In linear regression, the line of best fit is straight, as shown in the following diagram: The given data points are minimized by reducing residuals or offsets of each point from the line. Vertical offsets are generally used in surface, polynomial, and hyperplane problems, while perpendicular offsets are utilized in common practice. Depending on the following studies, the author used the least square method to optimize the model parameters by reducing errors.
https://doi.org/10.1007/s41024-020-00082-2
https://doi.org/10.1680/jgere.18.00014
https://doi.org/10.1007/s10706-013-9692-9
Ln 363-364 is not descriptions do not seem correct
Response:The information in lines 363-364 was checked and corrected.
Comments on the Quality of English Language
There are no significant comments on the quality of English.
Response:Thank you very much for your valid comments.